# Genetically diverse mice exhibit divergent domain-specific, sex-dependent behavioral outcomes following exposure to early life stress

Jennifer Nguyen[ID], Kevin Shimizu[ID], Varvara Zlotnik, Matthew N. Nguyen[ID], Sandra Del Toro[ID], Michael T. Nguyen[ID], Janet Ronquillo[ID], Lindsay R. Halladay[ID]*

Department of Neuroscience, University of Arizona, Tucson, Arizona, United States of America

* lrhalladay@arizona.edu

## Abstract

Understanding how genetic variability shapes responses to environmental and developmental factors is critical for advancing translational neuroscience. However, most preclinical studies rely on inbred mouse strains that do not capture the genetic complexity of human populations. One key area of translational research focuses on identifying the neural and behavioral consequences of early life trauma. Rodent models of childhood neglect, such as maternal separation with early weaning (MSEW), have been used in isogenic strains like C57BL/6J (B6) to identify behavioral domains and neural loci of deficits stemming from exposure to MSEW. To understand how genetic diversity may contribute to the outcomes produced by MSEW, and thus inform future studies on the topic, we utilized the Jackson Laboratory Diversity Outbred (DO) line, a population derived from eight founder strains that exhibit broad genetic and phenotypic heterogeneity. We first compared MSEW effects on social behavior in DO mice versus B6 mice, because we have previously found social behavior deficits in B6 mice with a history of MSEW. Indeed, we established that MSEW incited social motivation deficits in DO mice, in a sex-specific manner. We then expanded our investigation of DO mice to test MSEW-related changes in anxiety-like behavior, fear learning and expression, and reward-seeking. Results revealed that MSEW produces distinct, sex-specific phenotypes: female DO mice displayed reduced social motivation and elevated anxiety-like behavior, while male DO mice showed attenuated CS-evoked fear expression and diminished reward-seeking behavior. Additionally, immunohistochemical analysis revealed increased Fos expression in the paraventricular nucleus of the hypothalamus (PVN) in MSEW-exposed DO mice, both at baseline and following acute stress. These findings highlight the importance of considering genetically diverse models to better capture the nuances of early life adversity-related outcomes relevant to human populations.

**Data availability statement:** All relevant data are within the manuscript and its Supporting Information files.

**Funding:** L.R.H., The Jackson Laboratory Diversity Outbred Pilot Grant Program https://www.jax.org/ L.R.H., National Institute of Mental Health (NIMH) award R15MH127514 https://www.nimh.nih.gov/ Funders did not play a role in the study design, data collection, analysis, decision to publish, or preparation of the manuscript.

**Competing interests:** The authors have declared that no competing interests exist.

## Introduction

Early life adversity is a well-established risk factor for psychiatric disorders, not limited to generalized anxiety disorder, major depression, and impaired social functioning [1,2]. Rodent models of early life stress, such as maternal separation with early weaning (MSEW), have been instrumental in illuminating how exposure to stressors during sensitive developmental periods can shape behavioral and neurodevelopmental trajectories [3–6]. Much of the prior work in this area has used inbred mouse lines such as C57BL/6 (B6) mice to specifically control genetic influence, but being that isogenic strains are genetically uniform, this may severely limit the translatability of findings, especially since mouse behavior often varies widely across strain [7–9]. As such, use of genetically diverse animals may better capture the complex gene by environment interactions that underlie human susceptibility and resilience to early life adversity.

The Diversity Outbred (DO) mouse population, developed by the Jackson Laboratory [10,11], was designed to capture broad genetic variation by intercrossing eight founder strains to create offspring with extensive genetic and phenotypic heterogeneity. As such, DO mice may provide the opportunity to better approximate the behavioral and neural outcomes seen in humans, particularly in the context of complex developmental stressors. Early life adversity models such as MSEW have been unexplored in this population.

To remedy this, we first examined whether DO mice exhibit social behavior deficits following MSEW as we previously have observed in B6 mice [4,5], with the goal of validating whether the DO model captures similar core phenotypes. We then extended our investigation of DO mice to other behavioral domains also known to be altered by maternal separation procedures in inbred strains, including anxiety-like behavior [3,12,13], fear learning [14,15], and reward-seeking [16,17]. To assess underlying neural correlates in DO mice, we measured Fos expression in the paraventricular nucleus of the hypothalamus (PVN) in response to an acute stressor. In all, this evaluation of sex-specific and domain-specific outcomes following MSEW in a genetically diverse mouse population provides insight to whether genetic heterogeneity might offer greater relevance for understanding human outcomes than traditionally used isogenic strains.

## Materials and methods

### Animals

Adult male and female Jackson Laboratory Diversity Outbred mice ("DO," strain 009376, N = 309) and C57BL/6J mice ("B6," strain 000664, N = 49) were bred in our home vivarium (breeders were obtained from the Jackson Laboratory, Sacramento, CA, and are not counted in N above). In accordance with guidelines set by the Jackson Laboratory as part of the Diversity Outbred Pilot Grant Program, DO breeders provided were assigned to breeding pairs by the Jackson Laboratory to ensure maximum genetic heterogeneity, and, only one generation of DO mice was produced to reduce chances of genetic drift for that strain. After weaning, mice were housed

with same-sex littermates from the same experimental condition in a humidity- and temperature-controlled holding room on a 12:12h light/dark cycle (lights on at 0700). Cages housing non-stressed (NS) control mice were outfitted with a cotton nestlet and an acrylic enrichment tube [4,5] but mice that underwent maternal separation with early weaning (MSEW) were only provided a nestlet, but no acrylic tube, because environmental enrichment can in some cases reverse behavioral deficits related to early life adversity [12,18]. Experiments were approved by and carried out according to standards set by the Institutional Animal Care and Use Committee.

## Maternal separation with early weaning (MSEW)

Early life stress was modeled through MSEW, as we have previously described [4,5]. Briefly, half of the pups from each litter were randomly assigned to groups: MSEW (DO, n = 72 females and 87 males; B6, n = 12 females and 8 males) or NS (DO, n = 59 females and 91 males; B6, n = 19 females and 10 males). Pups' tails were colored with Sharpee to keep track of group assignment beginning on postnatal day (PD) 2. Pups assigned to the MSEW group were separated from the dam on PD 2 through PD 16 (4h/day on PD 2–5 and 8h/day on PD 6–16, during the light cycle) and weaned early on PD 17 (Fig 1). During the separation period, MSEW pups from each litter were placed together in a cage that included some of the nesting material from their home cage (each MSEW cohort was moved to the same cage each day for the 15-day period of separations). While separated, MSEW pup cages were placed on a heating pad to aid in thermoregulation, and experimenters checked on the pups throughout the day. No pups died during separations or experienced filial cannibalism upon return to the nest at the end of the day. NS pups were left undisturbed with the dam by experimenters until weaning on PD 23.

## Behavioral assays

At 10–12 weeks of age, mice were handled by experimenters for 2min/ day for 3 consecutive days prior to being tested behaviorally. Behavior was scored using an automated tracking system, EthoVision XT version 14 (Noldus, Leesburg, VA), which was calibrated (e.g., freezing thresholds) by experimenters blind to experimental conditions. On each test day, mice were acclimated to the behavioral testing room for at least 1h prior to being tested.

## 2-period social interaction (2pSI)

The 2pSI apparatus (Fig 2A) was a T-shaped, opaque acrylic maze consisting of three runways (30 × 7 cm), two of which included end-compartments (5 × 7 cm) that held a novel conspecific or object (TAP Plastics, San Jose, CA). The compartment enclosures were created by the insertion of transparent, air-holed acrylic barrier panels that enabled test mice to see, smell, and have limited tactile interaction with conspecifics. 2pSI was a 25-min session where test mice were first habituated to the maze for 5min (with no stimuli inside the compartments) then presented with inter-compartment stimuli for two consecutive 10-min social behavior phases (the test mouse freely explored the maze and was not removed or touched for the duration of

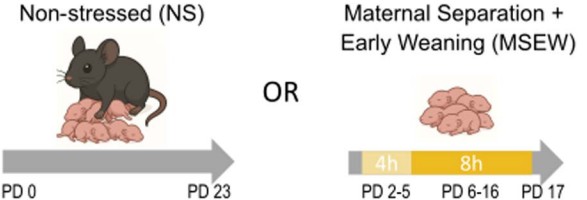

**Fig 1. Maternal separation with early weaning (MSEW).** Pups from each litter were randomly assigned to the non-stressed control (NS) or MSEW condition. Pups in the MSEW group were separated from the dam for 4 hours per day on postnatal days (PD) 2-5, and 8 hours per day on PD 6-16, and weaned on PD 17. NS pups were left undisturbed with the dam until weaning on PD 23.

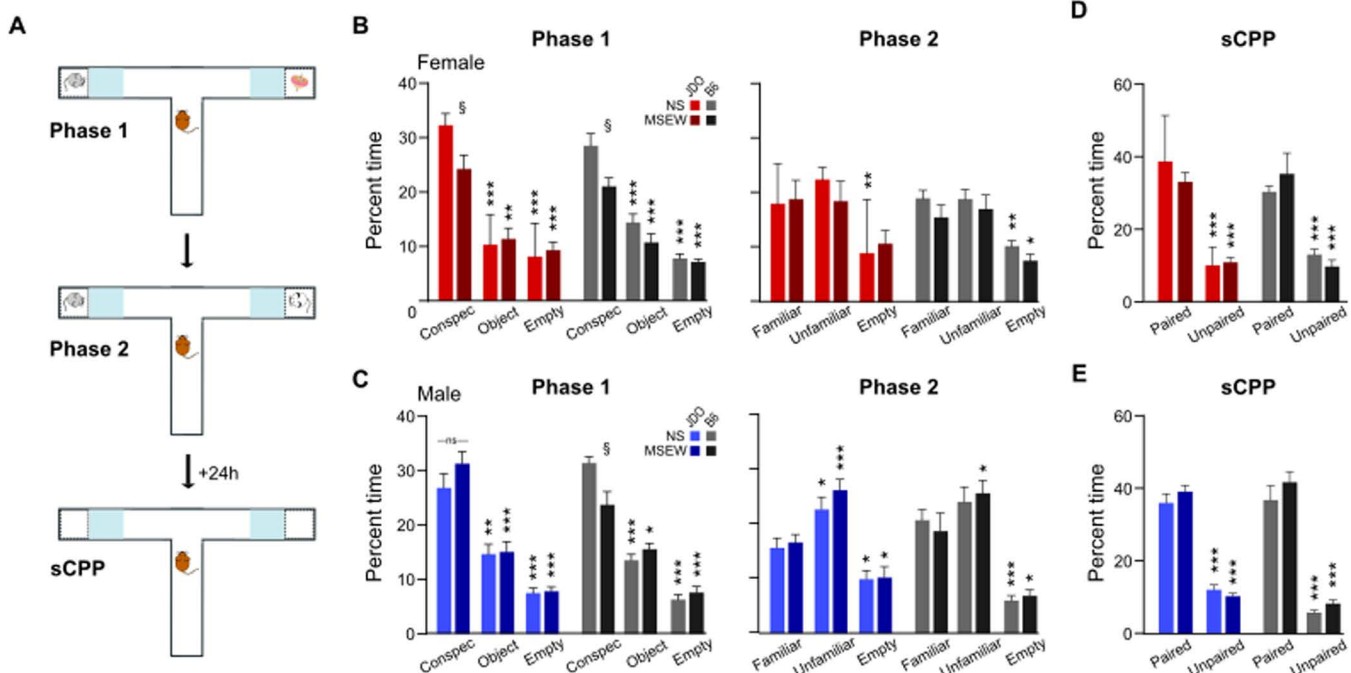

**Fig 2. Effects of MSEW on social behavior in DO and B6 mice.** (A) 2pSI procedures included a 5-min habituation phase when the test mouse could freely explore the empty maze, followed immediately by two consecutive 10-min phases to test social motivation (novel conspecific versus object) and social novelty preference (unfamiliar versus familiar conspecific). The next day, test mice were returned to the maze with no stimuli present for sCPP. (B) Female MSEW mice from both strains exhibited significantly attenuated social motivation compared to their NS counterparts during phase 1 of the 2pSI. During phase 2, DO (but not B6) MSEW females failed to spend more time interacting with social stimuli compared to the empty end of the maze. (C) In male mice, MSEW reduced social motivation in B6 but not DO groups, which may indicate resilience stemming from genetic diversity. (D, E) During sCPP, all groups exhibited a significant preference for locations previously paired with social stimuli compared to the empty end of the maze. Data are presented as mean ± SEM. Significant comparisons between stress conditions are denoted with '§' while comparisons of stimulus type within each experimental group are denoted by '*'. §/* = p < .05, ** = p < .01, *** = p < .001.

the session). During Phase 1, the test mouse could interact (through the barrier panels) with a same-sex novel conspecific or a novel object (left or right compartment, counterbalanced). During Phase 2, the novel object was replaced with another novel same-sex conspecific; the test mouse could interact with the original, "familiar" conspecific, or the new "unfamiliar" conspecific. Interaction was operationally defined as the time that the center-point of the test animal was tracked inside defined zones adjacent to the stimulus compartments. In the non-compartment end of the maze, an identical area was demarcated as the empty end zone, and time spent there was compared to the zones adjacent to stimuli. Between test mice, the apparatus was cleaned with an unscented disinfectant (Seventh Generation Inc., USA).

Novel conspecifics used in the 2pSI were bred in our lab but were unrelated to the test mice (i.e., they were not from the same dams birthing MSEW and NS pups). Each conspecific mouse was used up to three times per test day, and used for multiple non-consecutive test days to reduce the total number of animals used. (Often these animals were used for practice procedures or pilot experiments after being 'retired' from novel conspecific duties). These mice are not included in the reported N.

## Social conditioned place preference (sCPP)

sCPP took place 24h following 2pSI, in the same T-maze described above (no stimuli in the compartments), with the intention of quantifying social reward/ memory by determining whether test mice spent more time in the interaction zones

compared to the empty end zone. Test mice were allowed to explore the maze freely for 10 minutes. As both stimuli compartments contained conspecifics at some point during 2pSI, data were collapsed across stimuli zones and compared to the empty end zone. The apparatus was cleaned with an unscented disinfectant (Seventh Generation Inc., USA) between trials.

### 3-D radial arm maze (3DR)

The 3DR measures anxiety-like behavior and serves as an improved version of the elevated plus maze (EPM) [12] (Fig 3A). It is an opaque acrylic apparatus that consists of three zones of interest: a central octagonal hub (30 cm diameter, elevated 60 cm above the floor) connected to 8 upwardly angled (50°) mesh-covered bridges (15.2 × 11.2 cm) that then each connect to a horizontal arm (0°; 35 × 11.2 cm) that is positioned out of sight of a mouse located on the central platform (MazeEngineers, Skokie, IL). This configuration provides unfamiliar open spaces with no "safe" zones (e.g., closed

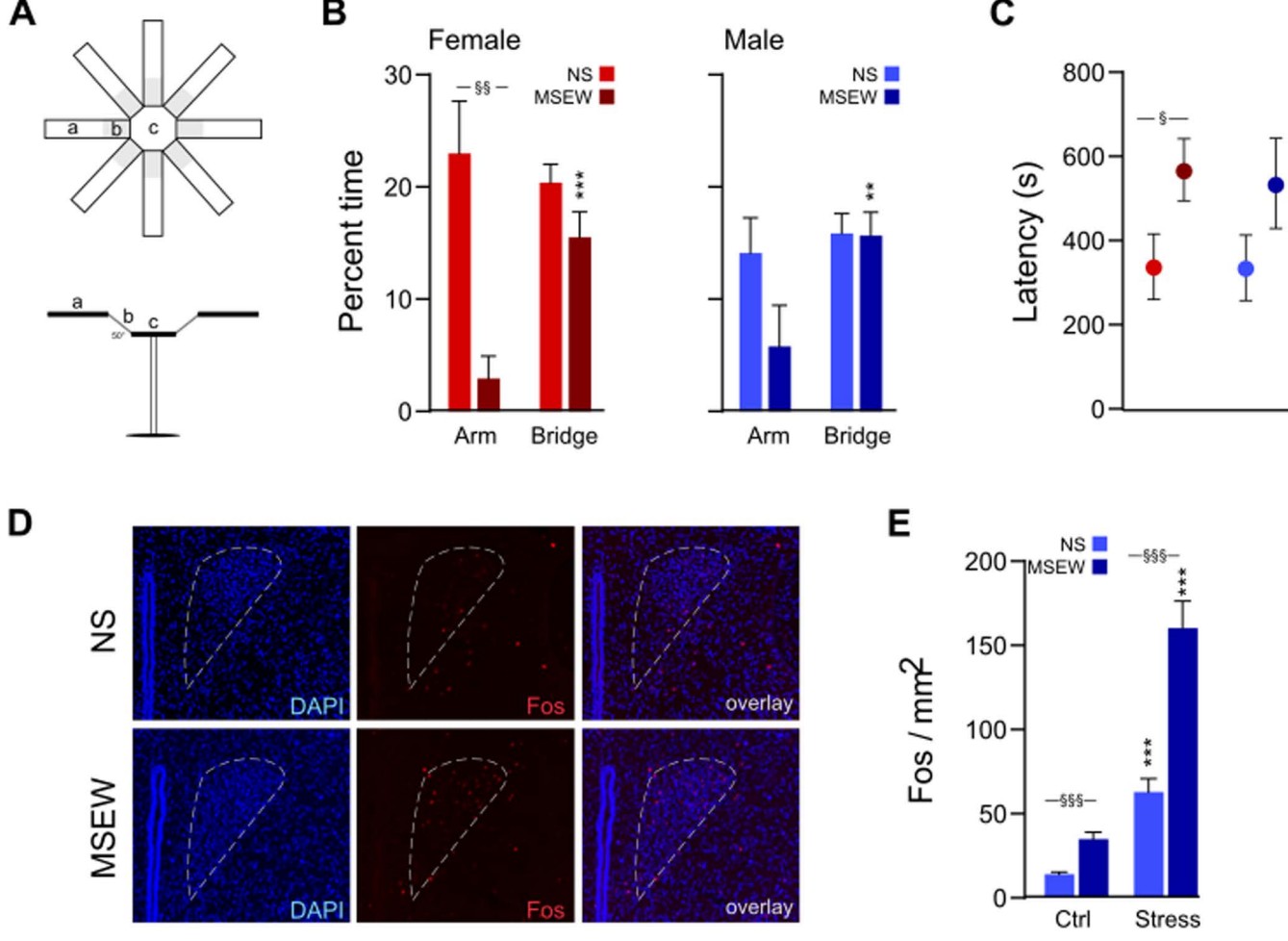

**Fig 3. MSEW effects on anxiety-like behavior and neural responsiveness.** (A) Exposure to MSEW significantly increased anxiety-like behavior only in female DO mice, as measured by (B) time spent in the arms of the 3DR and (C) latency to enter an arm of the 3DR. (D, E) Both basal Fos levels and FST-evoked Fos in the PVN were elevated in DO mice exposed to MSEW relative to NS groups. Data are presented as mean ± SEM. Significant comparisons between stress conditions are denoted with '§' while comparisons across stimuli for each experimental group are denoted by '*'. § = p < .05, §§ = p < .01, §§§/*** = p < .001.

arms of the EPM). Mice were placed on the central platform and allowed to freely explore the 3DR for 12 min. EthoVision XT recorded time spent in each zone (hub, bridges, or arms), as well as the latency to enter the arms of the maze. The 3DR was cleaned between mice with 70% EtOH.

### Fully-reinforced (FRF) and partially-reinforced (PRF) cued fear conditioning

Fear procedures took place in a 17 × 17 × 25 cm chamber with transparent acrylic walls and a metal rod floor for shock delivery (MazeEngineers, Skokie, IL). Acrylic wall and floor inserts allowed reconfiguration of the chamber to differentiate contexts A and B (transparent walls and grid floor or gray walls and solid floor, respectively). Chambers were configured to communicate with the behavioral tracking system, EthoVision XT version 14 (Noldus, Leesburg, VA). Fear conditioning was a 5-day procedure with each session occurring 24h apart. On day 1, mice were given 3 presentations of an auditory pure tone CS (30 sec, 3 kHz, 75 dB) to habituate them to the CS. On day 2, mice underwent fear conditioning (acquisition, context A, Fig 4A) consisting of a 3-min baseline period then either 3 presentations of the CS coterminating with a foot-shock US (2 sec, 0.6 mA), with variable ITIs between 60–120 sec (fully-reinforced fear conditioning; FRF), or 6 presentations of the CS, of which the first, third, and sixth presentation coterminated with a footshock, but not the second, fourth, or fifth CS (partially-reinforced fear conditioning; PRF). FRF and PRF groups experienced dissimilar acquisition days, but all subsequent days were the same for both groups. Day 3 consisted of an extinction session in context B, whereby after a 3-min baseline period, the CS was presented 50 times without a shock (10-sec ITI). On day 4, mice were returned to context B for a CS recall test, where they received 10 presentations of the CS alone. Then on day 5, mice were placed back in context A for a context (CX) recall test, which consisted of 10 minutes of exposure to the acquisition context, but without any tone or shock presentations. During each session, freezing behavior was tracked and quantified using EthoVision XT's 'immobility' measure, configured previously by our lab to correlate significantly with hand scoring of freezing behavior. In rare cases, some mice were able to jump out of the chamber despite 25 cm tall walls; those mice, which were not primarily from any one experimental group, were excluded from the entire study. Context A and B were cleaned between mice with 70% EtOH plus 1% vanilla extract or 1% acetic acid, respectively).

### Operant conditioning

To facilitate motivation to seek food rewards, mice underwent food restriction to decrease their body weight to 80–85% of their *ad lib*, free-feeding weight. Operant training took place in sound attenuated 27 × 27 × 11 cm chambers (Med Associates, Fairfax, VT), each equipped with two ultra-sensitive levers and a dispenser-receptacle system that delivered 20 mg grain-based dustless precision pellets (Bio-Serv, Flemington, NJ). Initially, mice were trained over consecutive days (24h between sessions) on a fixed-ratio (FR) 1 schedule until criterion was met (>35 rewards per session) as previously described [19,20]. Once an individual mouse met criterion, its schedule of reinforcement was changed to FR 3. After an individual mouse performed consistently at FR 3 (3 consecutive days of lever-pressing with <20% coefficient of variation), a 5-day period of extinction commenced where no rewards were given during the daily 40-min sessions when mice were allowed to freely lever-press for an unlimited number of trials. Testing for each mouse ended after the 5th extinction day.

### Immunohistochemistry

To assess the degree to which acute stress activates neurons in the paraventricular nucleus of the hypothalamus (PVN), NS and MSEW mice (n = 9 per stress condition) were either left in their home cage or subjected to a forced swim test, whereby mice were gently placed into a polypropylene cylindrical container (25 cm diameter × 30 cm height) 2/3 full with room temperature (21–23 °C) water, ensuring their heads did not become submerged. For 7 min, each test mouse was allowed to swim freely. Immediately at the end of the assay, an experimenter removed the test mouse and gently patted it dry with paper towel and placed it in a clean recovery cage, where it was left undisturbed for 90 minutes. After that time,

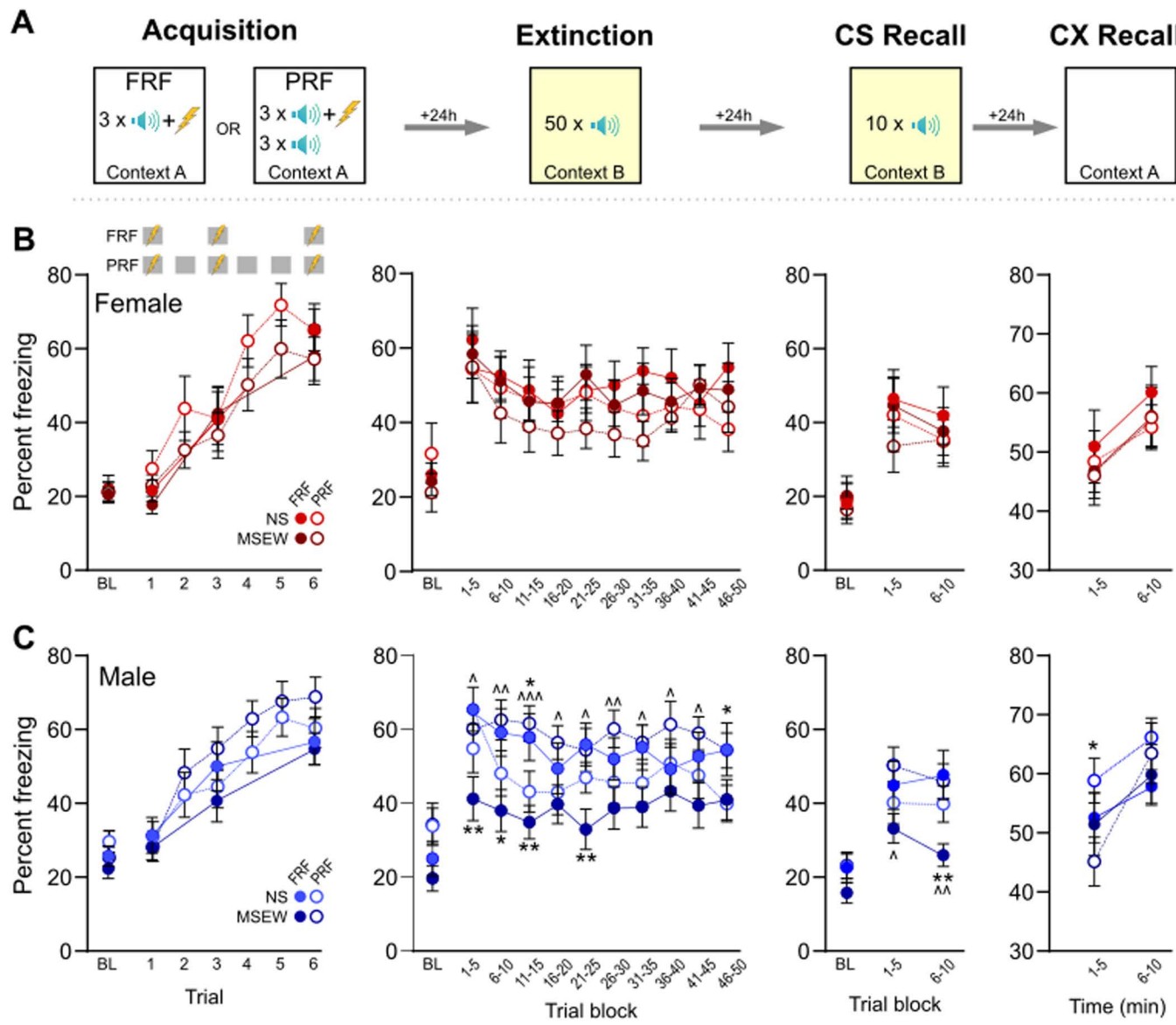

**Fig 4. Fear expression was altered in male DO mice exposed to MSEW.** (A) DO mice were trained on a fully reinforced (FRF) or partially reinforced (PRF) cued fear paradigm. (B) MSEW did not alter fear learning or expression in female DO mice. (C) MSEW reduced CS-evoked freezing in male DO mice trained on FRF during early extinction trials and the latter half of CS recall relative to NS mice. In mice trained on PRF, during some extinction trials MSEW elevated CS-evoked freezing and reduced freezing on CX recall day in male MSEW mice compared to NS mice. Data are presented as mean ± SEM. Comparisons between stress conditions are denoted with '*' while comparisons between protocols (FRF versus PRF in MSEW groups) are denoted by '^'. */^ = p < .05, **/^^ = p < .01, ^^^ = p < .001.

each mouse was given an overdose of Euthasol (150 mg/kg, IP injection; Henry Schein, Melville NY) and perfused intracardially with 4% paraformaldehyde (PFA). Brains were extracted and serial sections (30 μm) were taken from tissue containing the PVN.

Immunostaining took place over a 2-day procedure. On the first day, tissue was rinsed in PBS before incubation in blocking buffer with DAPI for 30 min, then incubated overnight in anti-c-Fos rabbit polyclonal primary antibody (Abcam

#ab190289) at room temperature. The next day, tissue was rinsed in PBS prior to incubation in goat anti-rabbit IgG polyclonal secondary antibody, Alexa Fluor 568 (Abcam #ab175471) for 1 hour. Tissue was then rinsed in PBS, mounted to gel-coated slides, and coverslipped with Fluoromount-G (Invitrogen, Carlsbad, CA). Slides were imaged at 20x using a Keyence BZ-X800 fluorescence microscope. Images were taken from tissue containing PVN, spanning ~ −0.58 to 0.94 mm A/P relative to Bregma and processed using ImageJ [21]. PVN area ($mm^2$) for each tissue section was measured using the polygon selection tool, with the aid of Paxinos and Franklin's mouse brain atlas. Cells within the selection region were counted manually by an experimenter blind to experimental conditions and divided by selection area to obtain a Fos+ per $mm^2$ for each tissue section (average 33 images per experimental group). Any images that contained damaged tissue (e.g., ripped or folded) or poor DAPI staining that prevented clear demarcation of the PVN were not included in the analysis.

### Statistical analysis

Statistical analyses were conducted using JASP (version 0.19.3) [22]. Behavioral and immunohistochemical data were analyzed using ANOVA, and when appropriate, post hoc tests or planned t-tests were used for pairwise comparisons. An alpha level of p = .05 was used for all analyses.

## Results

### Comparing social behavior in DO and B6 mice

Social motivation was assessed in Phase 1 of the 2pSI (Fig 2B,C left) by measuring time spent in the interaction zone of a novel, same-sex conspecific compared to time spent near a novel object or empty end. Separate three-way mixed ANOVA were used for each strain (stress × sex × stimulus). Overall, there was a main effect of stimulus for all groups (stimulus main effect, DO: $F_{(2,190)}=99.25$, p < .001, B6: $F_{(2,90)}=111.82$, p < .001). Planned comparisons revealed that within each strain, mice spent more time interacting with the novel conspecific compared to the novel object (DO: p < .001; B6: p < .001) or empty end (DO: p < .001; B6: p < .001), confirming that mice displayed social motivation. Additionally, mice spent less time in the empty end zone compared to the object zone, suggesting that mice also sought novelty in general relative to the absence of stimuli (DO: p < .001; B6: p < .001). In B6 mice, there was a main effect of stress ($F_{(1,45)}=6.93$, p = .01) because mice with a history of MSEW spent less time than NS mice in interaction zones, and a stress by stimulus interaction revealed that this was driven by a reduction in time spent interacting with the novel conspecific ($F_{(2,90)}=5.48$, p = .006). In DO mice, however, only female MSEW mice exhibited less time interacting with stimuli compared to NS mice (stress × sex interaction, $F_{(1,95)}=4.37$, p = .039) which was driven by interaction time with the conspecific (stress × sex × stimulus interaction, $F_{(2,190)}=3.12$, p = 0.47). Thus, during Phase 1, MSEW mice from both strains exhibited reduced social motivation, but in DO mice the effect was specific to females.

Social novelty preference was assessed in Phase 2 of the 2pSI by measuring time spent in the interaction zone of an unfamiliar same-sex conspecific (located in the compartment that previously held the object), or the familiar conspecific (located in the same compartment as in the prior phase) (Fig 2B,C right). Unlike Phase 1, exposure to MSEW did not significantly alter social behavior (stress main effect, DO: $F_{(1,95)}=0.26$, p = .61; B6: $F_{(1,45)}=1.03$, p = .32); time spent interacting with each stimulus was not dependent on prior exposure to stress for either strain (stress × stimuli interaction, DO: $F_{(2,190)}=0.06$, p = .94; B6: $F_{(2,90)}=0.39$, p = .68). In B6 mice, there was an interaction between sex and stimulus ($F_{(2,90)}=4.76$, p = .01) because males spent more time interacting with the unfamiliar conspecific compared to females (p = .04). This interaction between sex and stimulus was not present in DO mice ($F_{(2,190)}=1.38$, p = .25). Planned comparisons revealed that for male DO mice, NS and MSEW groups spent significantly more time with the unfamiliar conspecific than the familiar conspecific (NS: p = .02; MSEW: p = .002) and less time in the empty end (NS: p = .03; MSEW: p = .02). But for DO females, neither NS nor MSEW groups spent significantly different time with the unfamiliar conspecific (NS:

p=.14; MSEW: p=.95) compared to the familiar conspecific, and only NS females spent significantly less time in the empty end (NS: p=.002; MSEW: p=.08). This suggests that for DO mice, males, but not females, exhibited social novelty preference. For B6 mice, compared to the familiar conspecific, MSEW, but not NS males, spent significantly more time interacting with the unfamiliar conspecific (MSEW: p=.03; NS: p=.41), while both groups spent significantly less time in the empty end.

Twenty-four hours following 2pSI, test mice were returned to the maze for sCPP (Fig 2D,E). All mice spent more time in the zones previously associated with social stimuli compared to the empty end (stimulus main effect, DO: F(1,95)=303.83, p<.001; B6: F(1,44)=143.51, p<.001). Behavior in the sCPP was not affected by exposure to MSEW (stress main effect, DO: F(1,95)=0.42, p=.52; B6: F(1,44)=1.39, p=.25) or by sex (sex main effect, DO: F(1,95)=0.74, p=.39; B6: F(1,44)=0.26, p=.62). In DO mice, there was a trend toward significance for the interaction between all three factors (stress × sex × stimuli, F(1,95)=3.62, p=.06) because MSEW females spent less time in the zones that had been paired with social stimuli compared to other groups. This interaction was not significant in B6 mice (F(1,44)=.42, p=.52). Taken together, all mice exhibited a preference for spending time in zones that had been previous paired with social stimuli compared to the empty end of the maze.

## DO mice: anxiety-like behavior and associated neural activity

The 3DR was used to assess anxiety-like behavior by measuring time spent on the bridges and arms of the apparatus (Fig 3A). Data were analyzed using three-way mixed ANOVA (stress × sex × zone). There was an interaction between stress and zone (F(1,45)=9.36, p=.004) because MSEW mice spent significantly less time in the arms of the 3DR relative to NS groups, which was an effect driven by females (p=.002; Fig 3B). Additionally, while NS mice spent an equal amount of time in the arms and bridges (females: p=.61; males: p=.50), female and male MSEW mice spent significantly more time in the bridges than arms of the maze (females: p<.001; males: p=.007). Finally, female MSEW mice also exhibited greater latency to enter the arms than their NS counterparts (p=.05); this trend was true for male MSEW mice but the difference did not reach statistical significance (p=.17; Fig 3C). Taken together, these findings indicate that exposure to MSEW increased anxiety-like behavior.

Immunohistochemistry was carried out to assess whether the increase in anxiety-like behavior following MSEW could be due to differences in neural activity in the PVN (i.e., by expression of Fos following exposure to acute stress; Fig 3D). Data were analyzed using two-way ANOVA (stress × stimulus). There were significant main effects of both stress (F(1,128)=19.96, p<.001) and stimulus (F(1,128)=43.42, p<.001), as well as an interaction between the two (F(1,128)=8.35, p=.005) because exposure to acute swim stress increased Fos expression in NS mice and to a greater extent in MSEW mice, and MSEW mice also exhibited greater Fos levels at baseline (i.e., no acute stress exposure) than their NS counterparts (Fig 3E). Thus, in addition to behavioral effects, exposure to MSEW elevates baseline and acute stress-evoked activity in the PVN.

## DO mice: associative fear learning and expression

Fear learning and expression were assessed using separate three-way ANOVA with variables stated for each day below (Fig 4). On acquisition day, mice trained on both FRF and PRF paradigms exhibited significantly greater freezing to the CS compared to baseline (stimulus main effect, FRF: (F(3,156)=58.88, p<.001; PRF: (6,318)=54.0, p<.001) indicating that all groups acquired a fear response to the CS (Fig 4B,C). Behavior did not differ across experimental groups as a function of stress (FRF: F(1,52)=1.66, p=.66; PRF: F(1,53)=0.06, p=.81) or sex (FRF: F(1,52)=.20, p=.66; PRF: F(1,53)=1.19, p=.28), and there was no interaction between all three variables (FRF: F(3,156)=.70, p=.56; PRF: F(6,318)=.94, p=.46).

Freezing data from extinction day were analyzed using separate three-way ANOVA for each sex (stress × protocol × stimulus), and auditory stimuli were calculated in blocks of 5 CS presentations. In male mice, there was a significant

interaction between stress and protocol (F(1,63)=8.57, p=.005) because MSEW males trained on the FRF protocol exhibited less freezing than other groups. Planned comparisons of FRF data determined this was due to freezing during blocks 1–3 and 5 (p<.05). For the PRF groups, male MSEW mice froze more than NS mice only for blocks 3 and 10. And when comparing across protocols, male MSEW mice trained on FRF froze significantly less than male MSEW mice trained on PRF for each of the 10 blocks (p<.05), but there were no differences between FRF and PRF freezing among NS males. Unlike males, there were no group differences in female mice on extinction day (stress main effect, F(1,43)=.28, p=.60; protocol main effect, F(1,43)=1.16, p=.29; interaction, F(1,43)=.06, p=.80). Taken together, exposure to MSEW decreased freezing following a FRF paradigm, but only for male mice.

Freezing data from CS recall day were analyzed using separate three-way ANOVA (stress × protocol × stimulus), and auditory stimuli were calculated in blocks of 5 CS presentations. In male mice, there was a significant interaction between all factors (F(2,126)=3.17, p=.045). Similar to extinction day, MSEW males trained on the FRF protocol exhibited less freezing than other groups, which was more pronounced in the second block relative to the first block. Planned comparisons showed that for male MSEW mice trained on FRF also froze significantly less than MSEW males trained on PRF during both CS recall blocks (p<.05). There were no differences between FRF and PRF freezing among NS males. Finally, similar to extinction day, freezing did not differ across female experimental groups (stress main effect, F(1,42)=.21, p=.65; protocol main effect, F(1,42)=0.61, p=.44; interaction, F(1,42)=.05, p=.82) showing again that exposure to MSEW decreases conditioned fear expression but only in male mice.

Freezing data from CX recall day were analyzed using separate three-way ANOVA (stress × protocol × time). In all groups, there was a main effect of time because freezing increased between blocks 1 and 2 (male: (F(1,63)=34.97, p<.001; female: (F(1,43)=15.13, p<.001). In male mice, there was a significant interaction between stress and time (F(1,63)=5.04, p=.03) because MSEW mice exhibited a greater increase in freezing across time relative to NS males. Freezing did not differ in males as a function of protocol (F(1,63)=0.76, p=.39). For female mice, neither stress (F(1,43)=0.23, p=.63) nor protocol (F(1,43)=0.24, p=.63) affected freezing to the CX.

### DO mice: operant conditioning and extinction

Operant FR3 lever pressing data were analyzed using two-way ANOVA (stress × sex). There was a significant interaction between stress and sex (F(1,25)=6.44, p=.02), driven by a reduction in pressing in male MSEW mice compared to male NS mice (p<.05; Fig 5A). Lever pressing during extinction was analyzed using a mixed three-way ANOVA (stress × sex × session). There was a main effect of session (F(4,100)=50.80, p<.001) because on average mice reduced their lever pressing across sessions (Fig 5B). The interaction between all variables was significant (stress × sex × session, F(4,100)=8.87, p<.001) because male NS mice exhibited greater lever pressing rates than other groups. Planned comparisons revealed that this effect was driven by pressing during session 1 (p<.001), but not other days. Taken together, exposure to MSEW reduced reward seeking behavior relative to NS, but only in males.

### Discussion

These studies demonstrate that in DO mice, exposure to early life stress via MSEW alters multiple behavioral domains in a sex-dependent manner. The primary intent of this work was to explore whether introducing genetic heterogeneity affects whether exposure to MSEW alters aspects of social behavior, as we have reported previously in isogenic mice [4,5]. Indeed, we found that MSEW altered social motivation in both DO and B6 mice, validating that MSEW induces social behavior deficits regardless of genetic heterogeneity, with some caveats (e.g., sex-dependency) that we discuss in more detail below. Because MSEW introduced social behavior deficits in DO mice, and social behavior may depend on both aversive and rewarding properties [23], we conducted additional assays in DO mice to explore whether the social behavior effects observed here may stem from MSEW-induced alterations to aversive- or reward-related circuitry. Subsequently, through tests measuring anxiety-like behavior, fear learning, and reward-seeking, we gained important insight into

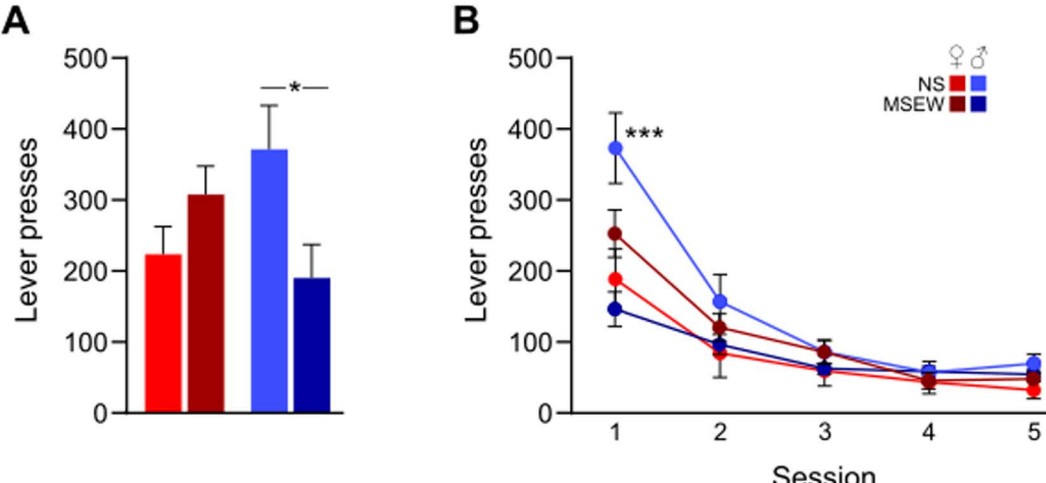

**Fig 5. MSEW attenuates reward-seeking behavior in male DO mice.** (A) In male DO mice, exposure to MSEW reduced the average rate of responding on a lever paired with food reward (FR 3) during 40-min sessions. (B) During an initial extinction session (day 1 of 5), MSEW reduced lever-pressing in male DO mice relative to male NS mice. Exposure to MSEW did not alter female DO reward-seeking. Data are presented as mean ± SEM. Significant comparisons between stress conditions for males are denoted with '*', * = p < .05, *** = p < .001.

the dynamics underlying MSEW-altered behavior and neural activity patterns in mice bred specifically for genetic diversity. These findings may inform the hypothesis that outbred populations, by virtue of their genetic complexity, may capture the variability and conditional effects of environmental factors observed in human populations [11,24].

### Social behavior

MSEW reduced social motivation for both sexes of B6 mice, as we have previously reported [4,5]. In contrast, for DO mice, only females exposed to MSEW showed a reduction in social motivation, which was apparent during both phases of the 2pSI: In phase 1, MSEW females spent less time interacting with the novel conspecific than NS females, while phase 2 revealed more pronounced deficits such that the female MSEW group failed to exhibit a preference for social stimuli over the empty end, whereas all other groups preferred the social stimuli. This effect was reminiscent of some of our prior work suggesting that mice with a history of MSEW exhibit habituation to social stimuli faster than NS groups [4]. It is important to note that prior studies utilizing mouse maternal separation have often produced disparate results likely due to variations in timing and duration of separations, as well as inclusion – or not – of female animals [6,25]. But here the inclusion of isogenic and DO mice of both sexes provides strong evidence that both sex and genetics may affect behavioral trajectories following exposure to early life stress via MSEW. These sex-specific findings may be indicative of a degree of resilience in male DO mice, perhaps due to factors reliant on genetic diversity [26–28].

The sCPP assay revealed no group differences in either strain, indicating that all groups preferred locations previously associated with social stimuli over non-social locations. This could suggest that MSEW may not alter the general rewarding nature of social interaction itself, but instead may impact real-time components of social behavior such as enhancing social vigilance or avoidance [25].

### Anxiety-like behavior and neural correlates of acute stress

Anxiety-like behavior, as measured in the 3DR, was increased following MSEW in DO mice, particularly among females. These results are consistent with our previous findings in inbred mice [4,12] and support the notion that exposure to MSEW enhances responsivity in stress-related circuits [29–32]. Indeed, we found increased Fos expression in the PVN of

DO mice exposed to MSEW, both at baseline and following an acute stressor, which endorses heightened neural activation in response to early-life adversity. These findings align with known sex differences in hypothalamic-pituitary-adrenal (HPA) axis reactivity [33–36] and further implicate the PVN in the behavioral consequences of exposure to early life stress.

### Cued fear learning and expression

Prior studies using maternal separation procedures have found impairments related to cued fear learning [14,15]. While MSEW did not alter the acquisition of a cued fear response, we found that MSEW selectively impaired CS recall in male, but not female, DO mice. Specifically, in MSEW males trained on FRF, freezing expression was decreased compared to NS males during CS recall (both early blocks of extinction and on CS recall day), in line with others' findings that maternal separation can impair cued fear recall [14,15] and thus may suggest altered amygdala function.

Because our recent work implicated the BNST in MSEW-altered behavior [5,6], in addition to employing a traditional FRF paradigm, we also ran cohorts of mice through a partially reinforced fear (PRF) paradigm to investigate fear expression in response to an ambiguous stimulus, which is known to be mediated by the BNST [37–39]. Interestingly, in mice trained on PRF, during some extinction blocks male MSEW mice exhibited greater freezing than male NS mice trained on PRF. This finding was more subtle than MSEW effects on FRF, but nonetheless suggests that MSEW may have altered BNST function to a degree in DO males.

One unexpected finding was the absence of protocol-specific differences in freezing that have been previously reported in inbred strains. B6 mice trained on PRF exhibit attenuated freezing upon CS recall compared to FRF conditioned groups [37]. In contrast, none of the groups tested here followed that pattern. Females from both stress conditions as well as NS males showed no difference in CS-evoked freezing regardless of whether trained on FRF or PRF protocols. Interestingly, male DO mice with a history of MSEW exhibited the opposite effect; those trained on FRF exhibited significantly less CS-evoked freezing than PRF-trained counterparts on both extinction and CS recall days. These results emphasize that genetic background may interact with environmental stressors and learning conditions to shape fear responses, but future studies would be warranted to determine this since for this set of experiments, we did not directly compare DO to inbred mice.

### Operant reward-seeking

During operant conditioning, all mice reached acquisition criteria for lever-pressing for a food reward. During subsequent days, MSEW reduced reward-seeking behavior in DO mice in a sex-dependent manner. Specifically, lever-pressing during FR 3 sessions, as well as during the first day of extinction, was decreased in MSEW males relative to NS males. This suggests that MSEW may reduce the motivation to seek food rewards, which is in line with other studies showing that rodent maternal separation can alter reward-seeking behaviors, with the caveat that prior work in this area is scarce and has generally found females to be affected to a greater extent than males [40,41]. This sex-specific discrepancy may again be attributed to the genetic heterogeneity of DO mice compared to other strains used in prior work, but additional studies are necessary to more thoroughly examine this possibility. Taken together, behavioral results from all experiments here suggest that MSEW disrupts both negative and positive valence systems of DO mice, in a sex-dependent manner.

### Sex-specificity of MSEW effects

We discovered an interesting divergence in MSEW effects on female versus male DO mice. MSEW primarily altered social motivation and anxiety-like behavior in DO females, but in males, MSEW mainly affected cued fear expression and reward-seeking. One possible explanation for these sex-dependent effects may relate to the distinct neonatal developmental trajectories of male and female mice. Neuroendocrine systems, including those regulating the HPA axis, undergo

sexually dimorphic maturation during early postnatal development [42,43] and are vulnerable to maternal separation [31,44]. Additionally, key neural systems implicated in the behaviors assayed here, including the BNST, amygdala, pre-frontal cortex, and nucleus accumbens undergo protracted and sex-specific maturation, and are sensitive to environmental insults during the postnatal timeframe that MSEW takes place [45–47], raising the possibility that variations in the developmental timeline between females and males account for the behavioral divergence observed here. For example, our results suggest that MSEW effects specific to female DO mice—namely reduced social motivation and increased anxiety-like behavior—may be mediated, at least in part, by developmental alterations in BNST circuits [6,48]. Conversely, MSEW-related impairments in cued fear recall and reward-seeking observed in DO males may reflect altered maturation of amygdala-based circuitry [49,50]. Importantly, these divergent trajectories may interact with genetic background to shape behavioral outcomes, highlighting the potential value of investigating whether genetically diverse models would better capture sex-specific windows of vulnerability.

Another potential contributor to the observed sex differences is that dams have been shown to respond more quickly to the ultrasonic vocalization (USV) distress calls of male pups compared to female pups [51]. As a result, upon return to the home cage each day, female pups may have experienced relatively longer periods of maternal absence, particularly during the first postnatal week when USV production is at its peak [52]. However, this possibility has yet to be directly tested.

## Conclusion

Taken together, these findings highlight the potential importance of using genetically diverse and developmentally sensitive models when investigating the long-term impact of early-life adversity. The nuanced and sex-specific effects observed in DO mice, particularly across distinct behavioral domains, seem to suggest that standard inbred models may miss key variability that is relevant to clinical application. These findings provide a framework for future mechanistic studies that consider how developmental timing and sex hormones interact with genetic background to influence the trajectory of early life stress-related outcomes.

## Supporting information

**S1 File. Data.** All experimental data are included in this file.
(XLSX)

## Author contributions

**Conceptualization:** Lindsay Halladay.

**Data curation:** Jennifer Nguyen, Kevin Shimizu, Varvara Zlotnik, Matthew N Nguyen, Sandra Del Toro.

**Formal analysis:** Kevin Shimizu, Lindsay Halladay.

**Funding acquisition:** Lindsay Halladay.

**Investigation:** Jennifer Nguyen, Kevin Shimizu, Varvara Zlotnik, Matthew N Nguyen, Sandra Del Toro, Michael T Nguyen, Janet Ronquillo.

**Methodology:** Lindsay Halladay.

**Project administration:** Lindsay Halladay.

**Resources:** Lindsay Halladay.

**Supervision:** Jennifer Nguyen, Lindsay Halladay.

**Writing – original draft:** Jennifer Nguyen, Lindsay Halladay.

**Writing – review & editing:** Jennifer Nguyen, Lindsay Halladay.

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
