## [Decision Letter · Decision Letter 0]

24 Jun 2025

PONE-D-25-27338Genetic diversity shapes behavioral outcomes and reveals sex differences in mice exposed to early life stressPLOS ONE

Dear Dr. Halladay,

Thank you for submitting your manuscript to PLOS ONE. After careful consideration, we feel that it has merit but does not fully meet PLOS ONE’s publication criteria as it currently stands. Therefore, we invite you to submit a revised version of the manuscript that addresses the points raised during the review process.

We look forward to receiving your revised manuscript.

Kind regards,

Alexandra Kavushansky, PhD

Academic Editor

PLOS ONE

Journal Requirements:

This work was supported by the Jackson Laboratory Diversity Outbred Pilot Grant Program and the National Institute of Mental Health (NIMH) award R15MH127514, awarded to LRH.

L.R.H., The Jackson Laboratory Diversity Outbred Pilot Grant Program

https://www.jax.org/

L.R.H., National Institute of Mental Health (NIMH) award R15MH127514

https://www.nimh.nih.gov/

Funders did not play a role in the study design, data collection, analysis, decision to publish, or preparation of the manuscript.

Reviewers' comments:

Reviewer's Responses to Questions

**Comments to the Author**

1. Is the manuscript technically sound, and do the data support the conclusions?

Reviewer #1: Partly

2. Has the statistical analysis been performed appropriately and rigorously? 

Reviewer #1: Yes

3. Have the authors made all data underlying the findings in their manuscript fully available?

Reviewer #1: Yes

4. Is the manuscript presented in an intelligible fashion and written in standard English?

Reviewer #1: Yes

5. Review Comments to the Author

Reviewer #1: The manuscript entitled ‘Genetic diversity shapes behavioral outcomes and reveals sex differences in mice exposed to early life stress’ by Nguyen et al. described the effect of maternal separation with early weaning (MSEW) on behaviors in the Diversity Outbred (DO) mouse strain. A series of behavioral battery tests showed the sex difference in anxiety and learning/memory.

The experimental designs are well organized, and the results are clear. However, the manuscript contains several points that require improvement.

Major comments

1. The title is misleading. Four behavioral assays were performed in the current study: two-period social interaction, 3-D radial arm maze, fear conditioning, and operant conditioning. However, the DO strain was compared with C57BL/6J only in the two-period social interaction test. Although the authors mentioned their previous reports using B6 mice in references 4 and 5 regarding social interaction, those studies employed different experimental conditions and cannot be directly compared to the results in the present study. Therefore, it is hard to claim that ‘Genetic diversity shapes behavioral outcomes and reveals sex differences…’. ‘Diversity Outbred mouse strain showed sex differences in behavioral outcomes caused by the exposure to early life stress’ seems more appropriate. The Discussion and conclusion should also be revised carefully.

2. The manuscript should be organized carefully. For example, Statistical analysis (Page 11, lines 251-254) must be included in the Materials & Methods section, not the Results section. Many figure legends are redundant and contain sentences to be included in the Materials & Methods or Results sections.

Overall, it is quite hard to conclude that genetic diversity affects behavioral outcomes and reveals sex differences in mice exposed to the MSEW. The manuscript should be carefully revised.

Minor comments

1. Are the error bars indicated as SD or SEM?

2. Symbols of significance between groups (*, **, or ***) are confusing in Fig 2, 3, and 5. Different symbols should be used like Fig. 4

6. PLOS authors have the option to publish the peer review history of their article (what does this mean? ). If published, this will include your full peer review and any attached files.

**Do you want your identity to be public for this peer review?** For information about this choice, including consent withdrawal, please see our Privacy Policy .

Reviewer #1: **Yes: ** Hitoshi Inada

---

## [Author Response · Author response to Decision Letter 1]

8 Aug 2025

Response to Editor Comments:

We have updated our formatting to adhere to PLOS ONE’s style requirements.

We have removed funding information from the Acknowledgment section. The Funding Statement (online submission) is accurate as is.

We have included a caption for our supporting information.

Response to Reviewers' comments:

Reviewer #1: The manuscript entitled ‘Genetic diversity shapes behavioral outcomes and reveals sex differences in mice exposed to early life stress’ by Nguyen et al. described the effect of maternal separation with early weaning (MSEW) on behaviors in the Diversity Outbred (DO) mouse strain. A series of behavioral battery tests showed the sex difference in anxiety and learning/memory.

The experimental designs are well organized, and the results are clear. However, the manuscript contains several points that require improvement.

Thank you for the positive feedback regarding experimental organization and clear results, and we appreciate your feedback related to improving the manuscript.

Major comments

1. The title is misleading. Four behavioral assays were performed in the current study: two-period social interaction, 3-D radial arm maze, fear conditioning, and operant conditioning. However, the DO strain was compared with C57BL/6J only in the two-period social interaction test. Although the authors mentioned their previous reports using B6 mice in references 4 and 5 regarding social interaction, those studies employed different experimental conditions and cannot be directly compared to the results in the present study. Therefore, it is hard to claim that ‘Genetic diversity shapes behavioral outcomes and reveals sex differences…’. ‘Diversity Outbred mouse strain showed sex differences in behavioral outcomes caused by the exposure to early life stress’ seems more appropriate. The Discussion and conclusion should also be revised carefully.

Thank you for this feedback. As requested, we have modified our title to avoid wording that alludes to a comparison between DO and isogenic mice. The updated title mentions only genetically diverse mice and summarizes the main findings relevant to the genetically diverse mice. It now reads: Genetically diverse mice exhibit divergent domain-specific, sex-dependent behavioral outcomes following exposure to early life stress

We appreciate the point related to our citing previous work (Refs 4 and 5), and we acknowledge that the social behavior testing apparatus we used here, a t-maze, is not physically identical to the apparatus we used in our past social behavior studies, which was more of an open field design. Nonetheless, the 2pSI is similarly intended to test both social motivation and social novelty preference, and in each case (Refs 4 and 5, as well as data presented here) social motivation (novel conspecific vs general novelty) was affected by MSEW. But because the tasks were indeed not identical, regarding the Introduction text that refers to 4 and 5, we removed the word “similar,” but retain the idea that each of these studies evidence an alteration to social motivation following exposure to MSEW.

2. The manuscript should be organized carefully. For example, Statistical analysis (Page 11, lines 251-254) must be included in the Materials & Methods section, not the Results section. Many figure legends are redundant and contain sentences to be included in the Materials & Methods or Results sections.

We appreciate this feedback. We have moved the Statistical Analysis section to Materials and Methods. Regarding the figure legends, after careful review, we removed text that may be overly redundant, but kept text that would enable a reader to better understand the figure without having to unnecessarily refer back to the methods / results. We feel that the legends are now more succinct while still containing information pertinent to the main findings of each figure.

Overall, it is quite hard to conclude that genetic diversity affects behavioral outcomes and reveals sex differences in mice exposed to the MSEW. The manuscript should be carefully revised.

We have carefully considered the statements in our manuscript and have made several changes throughout. For example, in the Discussion subsection “Cued fear learning and expression”, we made the following revision to remove (strikethrough) and replace text (underlined): These results emphasize that genetic background may interact with environmental stressors and learning conditions to shape fear responses, emphasizing the value of using genetically diverse animals alongside traditional inbred strains but future studies would be warranted to determine this since for this set of experiments, we did not directly compare DO to inbred mice.

Similarly in the subsection “Operant reward-seeking”, we added the following underlined statement: This sex-specific discrepancy may again be attributed to the genetic heterogeneity of DO mice compared to other strains used in prior work, but additional studies are necessary to more thoroughly examine this possibility.

Minor comments

1. Are the error bars indicated as SD or SEM?

Error bars depict SEM. We now specify that in the figure captions.

2. Symbols of significance between groups (*, **, or ***) are confusing in Fig 2, 3, and 5. Different symbols should be used like Fig. 4

Thank you for this feedback. We have updated Figures 2 and 3 with additional symbols that denote different comparisons, as well as the captions for figures 2, 3, and 5 to clearly define the comparisons being highlighted in each graph.

---

## [Decision Letter · Decision Letter 1]

17 Aug 2025

Genetically diverse mice exhibit divergent domain-specific, sex-dependent behavioral outcomes following exposure to early life stress

PONE-D-25-27338R1

Dear Dr. Halladay,

We’re pleased to inform you that your manuscript has been judged scientifically suitable for publication and will be formally accepted for publication once it meets all outstanding technical requirements.

Kind regards,

Alexandra Kavushansky, PhD

Academic Editor

PLOS ONE

Additional Editor Comments (optional):

Reviewers' comments:

Reviewer's Responses to Questions

**Comments to the Author**

1. If the authors have adequately addressed your comments raised in a previous round of review and you feel that this manuscript is now acceptable for publication, you may indicate that here to bypass the “Comments to the Author” section, enter your conflict of interest statement in the “Confidential to Editor” section, and submit your "Accept" recommendation.

Reviewer #1: All comments have been addressed

2. Is the manuscript technically sound, and do the data support the conclusions?

Reviewer #1: Yes

3. Has the statistical analysis been performed appropriately and rigorously? 

Reviewer #1: Yes

4. Have the authors made all data underlying the findings in their manuscript fully available?

Reviewer #1: Yes

5. Is the manuscript presented in an intelligible fashion and written in standard English?

Reviewer #1: Yes

6. Review Comments to the Author

Reviewer #1: (No Response)

7. PLOS authors have the option to publish the peer review history of their article (what does this mean? ). If published, this will include your full peer review and any attached files.

**Do you want your identity to be public for this peer review?** For information about this choice, including consent withdrawal, please see our Privacy Policy .

Reviewer #1: **Yes: ** Hitoshi Inada

---

## [Editor Report · Acceptance letter]

PONE-D-25-27338R1

PLOS ONE

Dear Dr. Halladay,

I'm pleased to inform you that your manuscript has been deemed suitable for publication in PLOS ONE. Congratulations! Your manuscript is now being handed over to our production team.

Kind regards,

on behalf of

Dr. Alexandra Kavushansky

Academic Editor

PLOS ONE